# Investigating Carvedilol's Repurposing for the Treatment of Non-Small Cell Lung Cancer via Aldehyde Dehydrogenase Activity Modulation in the Presence of β-Adrenergic Agonists

**Balqis A. Ikhmais** [1,*] **, Alaa M. Hammad** [1] **, Osama H. Abusara** [1] **, Lama Hamadneh** [2] **, Hamza Abumansour** [1] **, Qasem M. Abdallah** [3] **, Ali I. M. Ibrahim** [1] **, Lina Elsalem** [4] **, Mariam Awad** [1] **and Rahaf Alshehada** [1]

[1] Department of Pharmacy, Faculty of Pharmacy, Al-Zaytoonah University of Jordan, P.O. Box 130, Amman 11733, Jordan; alaa.hammad@zuj.edu.jo (A.M.H.); o.abusara@zuj.edu.jo (O.H.A.); h.abumansour@zuj.edu.jo (H.A.); a.ibrahim@zuj.edu.jo (A.I.M.I.); mariam7asan@hotmail.com (M.A.); rahafsaidghaith@gmail.com (R.A.)

[2] Department of Basic Medical Sciences, Faculty of Medicine, Al-Balqa Applied University, P.O. Box 206, Al-Salt 19117, Jordan; lama.hamadneh@bau.edu.jo

[3] Department of Pharmacology and Biomedical Sciences, Faculty of Pharmacy and Medical Sciences, University of Petra, P.O. Box 961343, Amman 11196, Jordan; qasem.abdallah@uop.edu.jo

[4] Department of Pharmacology, Faculty of Medicine, Jordan University of Science and Technology, P.O. Box 3030, Irbid 22110, Jordan; lmelsalem@just.edu.jo

[*] Correspondence: b.ikhmais@zuj.edu.jo

**Abstract:** Repurposing existing drugs appears to be a potential solution for addressing the challenges in the treatment of non-small cell lung cancer (NSCLC). β-adrenoceptor antagonist drugs (β-blockers) have tumor-inhibiting effects, making them promising candidates for potential NSCLC treatment. This study investigates the anticancer potential of a subset of β-blockers in NSCLC cell lines; A549 and H1299. Additionally, it investigates the underlying mechanism behind β-blockers' anticancer effect by influencing a potential novel target named aldehyde dehydrogenase (ALDH). The MTT assay assessed β-blockers' cytotoxicity on both cell lines, while Western blot and NADH fluorescence assays evaluated their influence on ALDH protein expression and activity. Carvedilol (CAR) was the most effective blocker in reducing cell survival of A549 and H1299 with $IC_{50}$ of 18 μM and 13.7 μM, respectively. Significantly, CAR led to a 50% reduction in ALDH expression and 80% decrease in ALDH activity in A549 cells, especially when combined with β-agonists, in comparison to the control. This effect might be attributed to β-agonist blockade or an alternative pathway. This novel finding adds to our understanding of CAR's multifaceted anticancer properties, implying that combining CAR with β-agonists could be a useful strategy for lung cancer treatment.

**Keywords:** repurposing; lung cancer; β-blockers; carvedilol; β-agonists; aldehyde dehydrogenase; NADH

## 1. Introduction

Lung cancer is the main cause of cancer-related deaths among men and women worldwide [1]. Lung cancer is categorized into small cell lung carcinoma (SCLC) and non-small cell lung carcinoma (NSCLC), with NSCLC constituting 80% of cases. Among NSCLCs, adenocarcinoma is the predominant subtype, representing 60% of cases [2]. It has recently been found that lung cancer contains cancer stem cells (CSCs), a subset of cells residing within the tumor mass. These cells are pivotal in driving tumor progression, heterogeneity, and resistance to treatment. Their presence contributes to the challenges of effectively treating and managing NSCLC, as they play a significant role in therapy resistance and potential disease recurrence [3,4]. Treatments for NSCLC include a variety of approaches, each with its own special mechanism of action. Growth factor receptor tyrosine kinase inhibitors (EGFR TKIs), which include the drugs gefitinib and erlotinib, have shown potential

in improving disease-free survival (DFS) for NSCLC patients with EGFR mutations. The introduction of osimertinib has been a breakthrough, greatly reducing the risk of cancer recurrence [5]. Clinical trials are currently evaluating the effectiveness of alectinib, a lymphoma kinase (ALK) inhibitor, for patients with ALK NSCLC [6]. Furthermore, dendritic cell- and chimeric antigen receptor (CAR) T cell-based immunotherapies are recently used treatments for NSCLC that act by targeting tumor antigens [7]. Despite the progress in treating NSCLC, the complexities of treatment resistance, tumor plasticity, and heterogeneity pose significant challenges. Therefore, apart from investigating new therapeutic agents, repurposing existing drugs has been considered to address this aggressive disease effectively [4]. Drug repurposing arises as a promising strategy due to its cost-efficiency and the expedited process of research and development [8]. This speed-up is facilitated by the accessibility of pharmacokinetic and pharmacodynamic data for the United States Food and Drug Administration (FDA)-approved drugs targeted for repurposing [4].

β-adrenoceptor antagonist drugs (β-blockers), a drug class effective in treating cardiovascular diseases, hyperthyroidism, migraines, and glaucoma, are currently undergoing expanded investigation for their potential anticancer properties within the context of drug repurposing [4]. The anticancer potential of β-blockers arises from their modulation of the sympathetic nervous system, which has been found to promote metastasis and cancer progression [9]. Solid tumors possess sympathetic nerves [10,11]. When triggered by persistent stress or depression, these nerves release endogenous catecholamines such as epinephrine (EPI) and norepinephrine (NE), which bind to α- and β-adrenoceptors within the tumor microenvironment [12–15]. Adrenoceptor activation influences critical pathways underlying cancer progression and metastasis [13]. Notably, β-adrenoceptor signaling is involved in various cancer-related cellular mechanisms [16], prompting researchers to propose that chronically used β-blockers could potentially impede cancer progression [17,18] and could be useful as adjuncts in cancer therapy [19–23]. Nilsson et al. have extensively reviewed preclinical and clinical data concerning β-adrenergic signaling in lung cancer [24]. This review substantiates the proposition of repurposing β-blockers for treating NSCLC [22].

Although there are several pieces of scientific evidence demonstrating the tumor-inhibiting effects of β-blockers [15,25,26], their precise anti-tumor mechanisms remain incompletely investigated [22]. Among β-blockers that might have potential in cancer treatment is carvedilol (CAR). Beyond the inhibitory effect of CAR on the β-signaling pathway, it possesses diverse attributes, such as antioxidative and antiproliferative properties [27–30], both of which are potential mechanisms for cancer inhibition [31–34]. Notably, CAR has been shown to diminish lipid peroxidation in human hearts, as evidenced by reduced levels of 4-hydroxy-2-nonenal (HNE) aldehyde and enhanced cardiac function, substantiating its antioxidant effects. This reduction might stem from various plausible direct and indirect detoxification mechanisms, potentially contributing to the protective role of β-blockers against cancer in the human body [35]. Furthermore, propranolol (PROP) has been discovered to reduce retinoic acid production, thereby impacting tumorigenesis and progression [36]. In addition, PROP inhibits acetaldehyde oxidation in ethanol-treated rats by inhibiting hepatic ALDH, with an efficacy of around 66% of disulfiram's inhibition [37]. Disulfiram, recognized as Antabuse, is a well-known irreversible inhibitor of aldehyde dehydrogenases (ALDH). It has been reported to influence the proliferation of various tumor cells, suppress cancer cell invasiveness, and prompt apoptosis through various in vitro cancer-related mechanisms [38]. N,N-diethylaminobenzaldehyde (DEAB) is also widely used as a reversible and minimally cytotoxic reference compound for ALDH inhibition [39].

The ALDH superfamily in humans comprises 19 isoenzymes within 11 families [40–42]. ALDHs, $NAD^+$-dependent enzymes, irreversibly oxidize internal and external aldehydes to their corresponding carboxylic acid [43], protecting organisms from oxidative stress [41]. They also play a pivotal role in metabolizing retinoic acid, essential for embryonic growth and epithelial differentiation [41,44]. Notably, ALDH expression and activity levels have been found to be heterogeneous among different types of solid tumors such as breast [45],

colorectal [46], lung [47], head-and-neck squamous cell carcinoma [48,49], prostate [50], pancreatic [51], bladder [52], and glioblastoma cancer [53]. Tumor heterogeneity was proposed to be attributed to stem-like cells expressing elevated ALDH levels. The elevated ALDH expression in CSCs is verified to induce tumor progression, metastasis, treatment resistance, and immune evasion [54–56], underscoring ALDHs as a remarkable CSC biomarker, notably in lung cancer [41,57]. Inhibiting ALDH isoenzymes or related pathways presents a promising therapeutic avenue for halting cancer progression, especially by targeting and eliminating CSC populations [41]. Flow cytometry of lung cancer cell lines and patient tumors indicated elevated ALDH activity in most NSCLCs, linked to ALDH1A1 expression [57]. Xenografts demonstrated 100-fold greater tumorigenicity in ALDH1A1-positive NSCLC cells when compared to ALDH1A1-negative cells. While ALDH1A1-positive tumors are considered malignant, the exact contribution of ALDH to NSCLC growth and survival requires further investigation [58]. Thus, in this study, we used NSCLC adenocarcinoma cell lines: A549 and H1299 cells expressing high and low levels of ALDH1A1, respectively [59,60]. Concerning other ALDH isoforms, ample evidence showed no expression of ALDH1A2 isozyme in H1299 cells [61], while other pieces of evidence showed low expression of ALDH in H1299 cells [62]. In addition, A549 cells have been found to express ALDH3A1 [63].

The current body of research concerning the anticancer effects of β-blockers on NSCLC is limited. Thus, in this study, the cytotoxicity of a subset of adrenoceptor blockers exhibiting varying selectivity toward adrenoceptors was examined against NSCLC adenocarcinoma cell lines A549 and H1299, cells expressing high and low levels of ALDH1A1, respectively [59,60]. These cell lines were then subjected to CAR or PROP as a single agent and combined with β-agonists to investigate their hypothesized ALDH-mediated anticancer action in NSCLC.

## 2. Results

### 2.1. Measuring the Basal Level of ALDH in A549 and H1299 Cell Lines

The basal activity level of ALDH in A549 and H1299 cell lines was measured using NADH fluorescence spectrophotometric assay. As shown in Figure 1A, A549 cells express highly active ALDH compared to H1299 cells, which show a very low activity level of ALDH. In order to confirm these findings, the basal protein level of a specific isoform of ALDH, which is ALDH1A1, was measured using Western blot analysis. A549 cells manifest a higher protein expression level of ALDH1A1 compared to H1299 cells, as shown in Figure 1B.

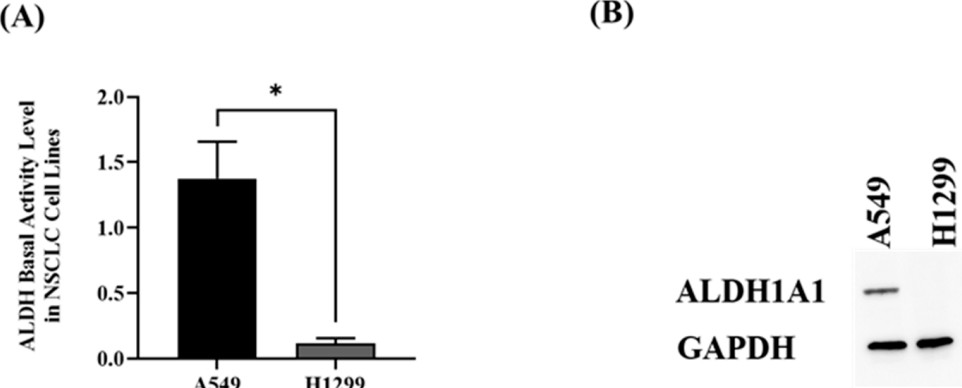

**Figure 1.** (**A**) ALDH basal activity for the lysates prepared from A549 (black column) and H1299 (grey column) cell lines was measured via fluorescence spectrophotometric assay; *n* = 3; * *p*-value < 0.05. (**B**) ALDH1A1 basal protein expression in A549 and H1299 cell lysate was measured using Western blot along with GAPDH as loading control.

### 2.2. Measuring Cytotoxicity of Drugs in A549 and H1299 Cell Lines

MTT colorimetric assay was carried out to measure the cytotoxicity of DEAB, β-adrenergic agonists (ISO and EPI), and commonly used β-blockers (including a subset of non-cardioselective, cardioselective β-blockers and mixed α/β-blocker) in A549 and H1299 cell lines over different ranges of concentration following 96 h exposure time. The $IC_{50}$ value of each compound was extracted from its dose–response curve and listed in Table 1. Dose–response curves for the β-blockers are presented in Figure 2. Curves of the rest of the compounds are also represented (Supplementary Figure S1).

**Table 1.** Toxicity of DEAB, β-agonists (ISO and EPI) and β-blockers on NSCLC cell lines (A549 and H1299). Cells were treated for 96 h; $n = 3$.

| Name of Compounds | Toxicity ($IC_{50}$ (μM) ± SE) in NSCLC Cell Lines at 96 h | |
| --- | --- | --- |
| | **A549** | **H1299** |
| DEAB | >100 | >100 |
| Isoproterenol | 88.3 ± 6 | 38 ± 4 |
| Epinephrine | 78.7 ± 2.3 | 23.5 ± 3.3 |
| Atenolol | >1000 | >1000 |
| Esmolol | 513.3 ± 51.7 | 400 ± 30.6 |
| Nadolol | >1000 | >1000 |
| Metoprolol | 706.7 ± 31.8 | 570 ± 60.3 |
| Bisoprolol | 730.0 ± 79.4 | 560 ± 112.4 |
| Propranolol | 146.3 ± 29.3 | 76 ± 13.3 |
| Carvedilol | 18 ± 2.1 | 13.7 ± 0.3 |

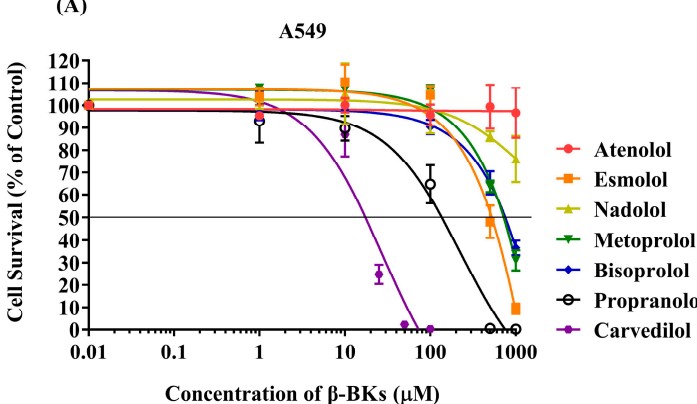

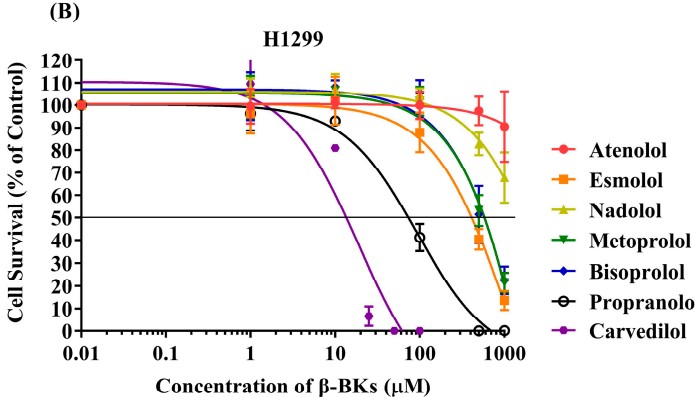

**Figure 2.** Dose–response curves for testing non-cardio- and cardio-selective β-blockers on NSCLC cell lines ((**A**) A549 and (**B**) H1299) as measured by the MTT assay. Treatment for 96 h; $n = 3$.

The compounds' cytotoxicity was found to increase in a dose-dependent manner. CAR, followed by PROP, demonstrated the highest cytotoxicity against A549 and H1299 cells, with $IC_{50}$ values of 18 μM and 13.7 μM for CAR and 146.3 μM and 76 μM for PROP, respectively, compared to the other tested β-blockers, as seen in Table 1. In contrast, esmolol, metoprolol, and bisoprolol showed mild cytotoxicity in both cell lines, with $IC_{50}$ values ranging from 500 μM to 1000 μM, while atenolol and nadolol showed the least cytotoxicity even at a high concentration of 1000 μM. EPI and ISO demonstrated comparable $IC_{50}$ values for each cell line, being 78.7 μM and 88.3 μM in A549 cells and 23.5 μM and 38 μM in H1299 cells, respectively. Notably, combining CAR (10 μM) with ISO (20, 40, and 60 μM) resulted in more reduction in the cell survival of A549 and H1299 compared to the control or single treatment, as shown in Supplementary Figure S2. DEAB exhibited no cytotoxicity within the studied concentration range. Therefore, a high non-toxic concentration of 80 μM DEAB was used alone and combined with other compounds for comparison assays.

### 2.3. Assessing ALDH Activity Level following Exposure to Different Treatment Conditions

NADH fluorescence spectrophotometric assay was used to assess the effects of CAR and β-adrenergic agonists (ISO and EPI) on ALDH activity levels in A549 cells under various treatment conditions. These conditions included CAR alone at 10 μM, ISO alone at 20, 40, and 60 μM, EPI alone at 12.5, 25, and 50 μM and a combination of CAR (10 μM) with either ISO at 20, 40, and 60 μM or EPI at 12.5, 25 and 50 μM concentrations. β-adrenergic agonists were introduced because β-blockers have shown the strongest impact when catecholamine levels are elevated [64]. The A549 cells were treated for 120 h, during which the protein expression occurred, and changes in its levels could be detected. H1299 cell data were excluded since these cells exhibited undetectable ALDH activity both in the untreated state and under the above treatment conditions.

It is clear in Figure 3A that monotherapies of either CAR or ISO cause no changes in the ALDH activity levels at the concentrations above. However, gradient reductions in ALDH activity levels were observed when CAR at 10 μM was combined with a range of concentrations of ISO, being the most significant ($p$-value < 0.0001) with approximately 80% reduction when combined with the highest concentration of ISO (60 μM) as compared to untreated control and to monotherapy-treated samples. Similar observations have been made with EPI, except that it was less efficient than ISO in reducing the ALDH activity, with only a 50% reduction in ALDH activity when combining 50 μM of EPI with 10 μM of CAR, as shown in Figure 3B. Interestingly, the efficiency of the combined treatment of CAR (10 μM) plus ISO (60 μM) in the inhibition ALDH activity was higher than that achieved by DEAB, with an approximately 20% difference.

To further prove that the ALDH is the putative target of CAR and ISO actions, the same cell lysates prepared from the combination-treated samples were exposed to DEAB at 80 μM. This concentration was selected based on an optimization experiment for the DEAB concentration to be non-toxic and efficiently inhibit ALDH activity with approximately 60% reduction compared with the untreated control, as shown in Figure 3. It was found that DEAB was highly efficient at inhibiting ALDH activity in the samples pretreated with the low concentration of combination therapies of CAR 10 μM plus ISO 20 μM or plus ISO 40 μM, with 53% and 55% reduction in ALDH activity when compared to the combination-treated samples without adding DEAB, respectively. Comparable results were achieved when A549 cells were pre-treated with a low concentration of EPI alongside CAR subjected to DEAB. This demonstrates that DEAB efficiency in inhibiting ALDH was only in the samples expressing ALDH and that CAR can affect ALDH activity exclusively in the presence of adequate concentrations of catecholamines (i.e., 60 μM of ISO and 50 μM of EPI), as seen in Figure 3B. Non-detectable ALDH activity was found in H1299 cell samples treated under the above treatment conditions.

Notably, PROP was investigated both individually and in combination with ISO. However, neither PROP alone nor in combination with ISO could diminish ALDH activity

in samples treated for 120 h, even at high concentrations. This underscores that the substantial changes in ALDH activity were primarily attributed to CAR's actions.

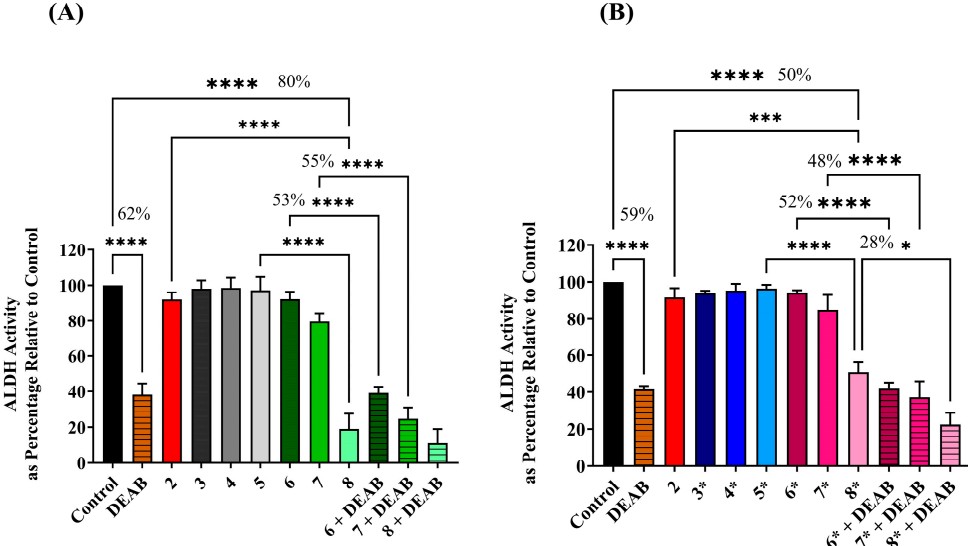

**Figure 3.** ALDH activity in A549 cells was measured under different treatment conditions using NADH fluorescence spectrophotometric assay. The percentages centered over the scales indicate the reduction level of ALDH in comparison between samples. (**A**) The figure refers to the A549 cells treated with ISO at various single concentrations, including 20, 40, and 60 μM, which refer to numbers 3, 4, and 5, respectively, and in combination with 10 μM of CAR as referred to by numbers 6, 7, and 8, respectively. Bar graph of DEAB generated from adding DEAB at 80 μM concentration to untreated cell lysate. Combination pretreated samples exposed to DEAB at 80 μM concentration represented as 6, 7, and 8 plus DEAB. (**B**) Star over the numbers on the x-axis indicates that EPI was used instead of ISO at concentrations of 12.5, 25, and 50 μM and represented as 3\*, 4\*, and 5\*, respectively. Combinations of 10 μM CAR with the above concentrations of EPI are represented as 6\*, 7\*, and 8\*, respectively. Number 2 is common between the two figures and refers to CAR studied at 10 μM concentration. (\* $p \leq 0.05$, \*\*\* $p \leq 0.001$, and \*\*\*\* $p \leq 0.0001$).

### 2.4. Assessing ALDH Protein Expression Level following Exposure to Various Treatment Conditions

To comprehend the mechanism behind the modified ALDH activity levels, the ALDH1A1 isoform was quantified using Western blot technique under the same treatment conditions. The selection of ALDH1A1 was based on the literature demonstrating that ALDH1A1 is highly expressed in A549 cells [38,59,63]. The cell lysates prepared for the activity assay were also used to investigate changes in ALDH1A1 protein levels. The resulting blots were incubated with ALDH1A1 antibody and analyzed, as seen in Figure 4. Despite H1299 cells demonstrating minimal or absent ALDH1A1 protein expression [65], they underwent the same treatment conditions as A549 cells.

The blot in Figure 4A reveals that individually treating A549 cells with CAR at 10 μM concentration or ISO at 20, 40, and 60 μM concentration did not influence the ALDH1A1 protein level. However, combined treatment of CAR and ISO led to a gradual reduction in ALDH1A1 protein expression, with the most significant reduction of around 50% observed at CAR 10 μM plus ISO 60 μM concentration ($p < 0.001$) and ($p < 0.05$) compared to control and ISO 60 μM treatment, respectively. These findings align with the activity assay, explaining the altered ALDH activity with CAR and ISO combination therapy. Similar reductions in ALDH1A1 protein expression were observed in A549 samples treated with combined CAR and EPI therapy, as seen in (Figure 5A). Notably, this reduction followed a concentration gradient, where higher EPI concentrations (12.5, 25, and 50 μM) increased significance in the reduction of ALDH protein compared to untreated control. In parallel, H1299 cells treated under identical conditions showed no detectable ALDH1A1 protein in

both control and treated samples, confirming the lack of activity in the NADH fluorescence assay results, as seen in Figures 4C and 5C. It is worth mentioning that treating A549 cells with 80 μM DEAB for 120 h had no impact on ALDH1A1 protein expression or activity.

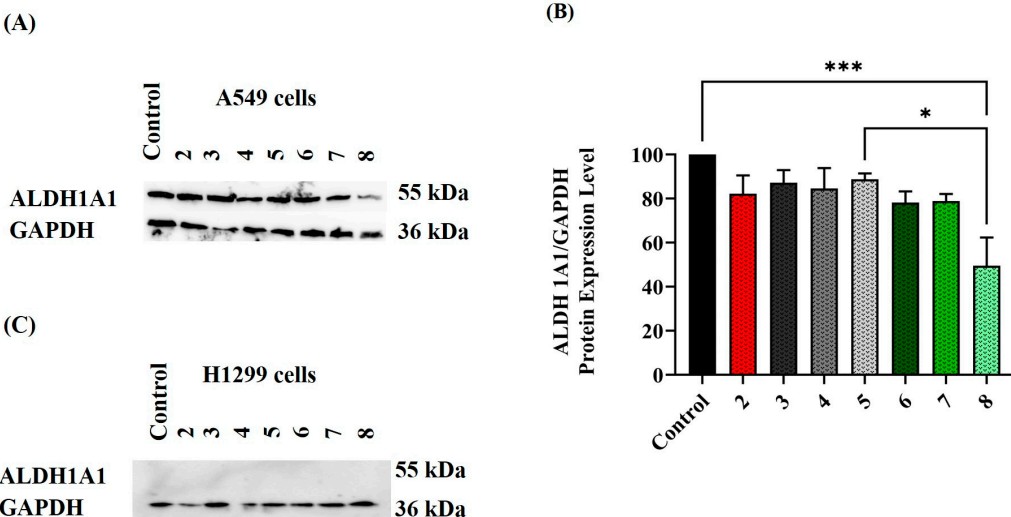

**Figure 4.** (**A**) ALDH1A1 protein expression in 120 h-treated A549 cells as measured by Western blot technique. The treatment conditions used were numbered as follows: 2, 3, 4, 5, 6, 7, and 8, which indicated A549 cells treated with CAR at 10 μM, ISO at 20, 40, and 60 μM and CAR 10 μM plus ISO 20, 40 and 60 μM concentrations, respectively. (**B**) Densitometric analysis was performed on three independent blots to measure the intensity of all bands. ALDH1A1 bands were first normalized to GAPDH; the means of these values were then represented graphically with ($\pm$ SE). (**C**) The H1299 cell lysate was prepared from the same treatment conditions used for A549 cells. One-way ANOVA was used to analyze the data, * $p$-value < 0.05 and *** $p$-value < 0.001.

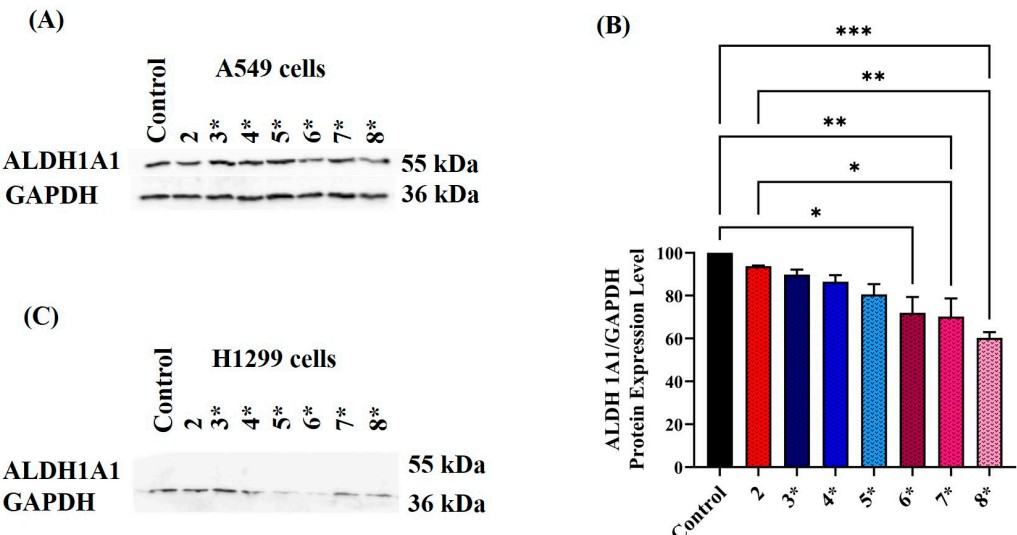

**Figure 5.** (**A**) ALDH1A1 protein expression in 120 h-treated A549 cell as measured by Western blot technique. A549 cells were exposed to the same treatment conditions mentioned earlier, except for using EPI instead of ISO. 2, 3*, 4*, 5*, 6*, 7*, and 8* indicate the cells treated with CAR at 10 μM concentration, EPI at 12.5, 25 and 50 μM concentrations and CAR 10 μM plus EPI 12.5, 25, and 50 μM concentrations, respectively. (**B**) Densitometric analysis for the blots is represented as a means of normalized ALDH1A1 value relative to GAPDH ($\pm$SE). (**C**) Blot for H1299 cells exposed to the same treatment conditions used for A549 cells. One-way ANOVA was used to analyze the data (* $p$-value < 0.05, ** $p$-value < 0.01 and *** $p$-value < 0.001).

## 2.5. Assessing ALDH Gene Expression Level upon Exposure to Various Treatment Conditions

To more deeply understand the molecular mechanism underlying the changes in ALDH activity upon prolonged exposure to β-adrenergic agonists and antagonists, it was necessary to measure the gene expression of ALDH1A1and β1- and β2-adrenoceptors using real-time PCR.

Notably, neither individual nor combined therapies significantly affected ALDH1A1 gene expression. However, significant upregulation was observed in β-adrenoceptors, particularly β2-adrenoceptor with a remarkable 11-fold increase under combined CAR 10 μM and ISO 60 μM therapy compared to control, as seen in Figure 6C. Interestingly, α-adrenoceptor gene expression was not detected in A549 and H1299 cells.

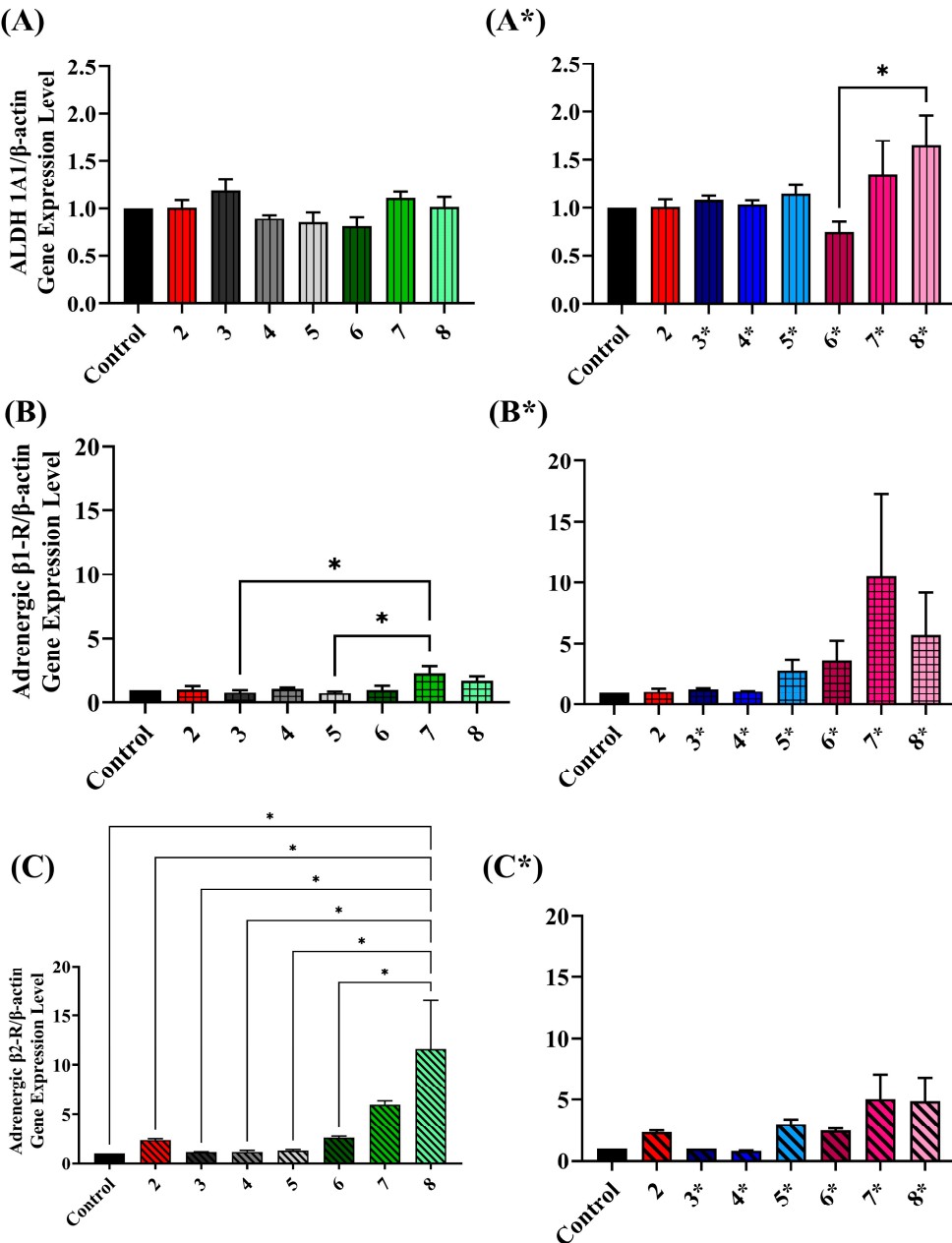

**Figure 6.** Fold change in gene expression level of ALDH1A1, β1- and β2-adrenoceptors was calculated relative to control from three independent experiments and represented as mean (±SE) in figures (**A,A\***), (**B,B\***) and (**C,C\***), respectively. A549 cells were treated for 96 h under various treatment conditions. The numbers in these figures are indication similar to those mentioned in Figures 4 and 5. One-way ANOVA was used to measure the level of significance, represented as a single asterisk with a $p$-value < 0.05.

### 3. Discussion

With the rise in drug repurposing, the potential anticancer action of β-blockers is being extensively investigated [4]. Nevertheless, uncertainties about β-blockers' efficacy in inhibiting cancer progression and their mechanisms persist [18]. Therefore, the anticancer action of a subset of β-blockers was examined in A549 and H1299 NSCLC adenocarcinoma cell lines, expressing both β1- and β2-adrenoceptors [66,67]. Our study revealed CAR as the most effective drug in reducing cell survival, followed by PROP, compared with the tested β-blockers, in both cell lines. Despite CAR being a mixed α/β-blocker, based on our results, its action may be mediated through β-adrenoceptor inhibition, particularly β2-adrenoceptor, as real-time PCR analysis demonstrated the absence of α-adrenoceptor in A549 and H1299 cells. This was also supported by our findings that PROP, the non-selective β-blocker, was effective at lower concentrations, whereas selective β1-blockers showed efficacy only at higher concentrations at which they become less selective, targeting β2-adrenoceptor as well [68]. According to previous data in the literature, the study by Molenaar et al. demonstrated that CAR has higher efficacy in blocking human β2-adrenoceptors than β1-adrenoceptors [69]. To the best of our knowledge, this is the first study demonstrating the potential anticancer action of CAR and PROP in A549 and H1299, established human NSCLC cell lines. CAR has demonstrated inhibition of cancer cell proliferation, apoptosis induction, and reduced invasiveness in breast cancer [70]. Similarly, PROP has shown the potential to reduce proliferation, induce apoptosis, and improve patient outcomes in breast cancer cases [71]. Nonselective β-blockers, including CAR and PROP, have been suggested for efficiently inhibiting tumor progression over selective β1-blockers [72].

Our research highlights the clinical significance of using β-blockers to treat chronic disease in cancer patients, enabling them to simultaneously manage their chronic condition and inhibit tumor growth. The favorable pharmacokinetic and pharmacodynamic profile, accessibility, and affordability of β-blockers also potentiate the significance of our findings. Ongoing research is necessary to improve treatment outcomes and combat resistance problems in NSCLC. Exploring novel cancer survival pathways and drugs, such as those targeting the ALDH pathway, holds promise in overcoming resistance to current lung cancer treatments. This research has the potential to revolutionize NSCLC management by offering more effective strategies to combat the disease.

This study showed how β-blockers along with β-agonists may contribute to the anti-cancer effects in NSCLC. Several experiments have been conducted to elucidate further CAR's underlying anticancer mechanism, a topic lacking a clear explanation in the existing literature [22]. Beyond β-adrenoceptor blockade, CAR has various roles, including antioxidant and antiproliferative activities, contributing to its anticancer effects [34]. We hypothesized that CAR's anticancer action might involve targeting ALDH, a detoxifying enzyme that mitigates oxidative stress, leading to therapeutic resistance in many solid tumors [41,73,74]. Genetic knockdown or pharmacological inhibition of ALDH1A1 in lung cancer enhances oxidative stress and sensitivity to chemotherapy [41]. We tested this hypothesis by exposing NSCLC cells to prolonged CAR and β-agonist treatment. Our findings showed CAR effectively reduced ALDH protein expression and activity when combined with β-agonists (combined-treated samples). To further confirm CAR's targeting of ALDH, we applied DEAB to combined-treatment samples. DEAB effectively targeted ALDH only in samples with low-concentration combination therapy, where significant ALDH level changes did not occur due to mild β-agonist concentrations. In contrast, DEAB's efficacy was limited in samples treated with high-concentration combined treatment of CAR and ISO, where ALDH levels were already low. Reduction in ALDH was seen in protein expression and activity rather than gene expression, suggesting post-transcriptional and translational effects of the CAR and β-agonist combination. This aligns with Yen et al.'s study on protein-mRNA level disparities, emphasizing that the regulation of mRNA post-transcriptionally can lead to differences and that mRNA presence does not ensure translation or protein function [75]. In addition, Qian et al. believed that results for the

protein level were more reliable and instructive than mRNA [76]. Thus, CAR's actions on ALDH could be attributed to ISO and EPI blockade or an adrenoceptor-independent pathway [34]. CAR's multifunctionality, beyond β-adrenoceptor blocking, underlies its observed anticancer action in this study and previous in vivo and clinical studies [77,78], potentially stemming from simultaneous inhibition of multiple oncogenic mechanisms [34] in addition to ALDH.

The lack of ALDH changes in PROP and β-agonist combined treatment samples indicated that ALDH was not involved in PROP's cytotoxicity. Several studies suggest PROP's anticancer mechanism involves decreasing pro-proliferative Ki-67 and pro-survival Bcl-2 markers, disrupting cell cycle progression, altering cyclin levels, elevating p53, enhancing caspase cleavage, and inducing apoptosis, as demonstrated in breast cancer cases and other tumor types [79–84].

In summary, CAR and PROP exhibit the strongest inhibition of cell survival in NSCLC cells compared to other tested β-blockers. While CAR's mechanism of cancer inhibition, involving ALDH inhibition, is a novel discovery, as presented in Figure 7, other pathways may also contribute, as seen in H1299 cells, which lack ALDH yet display cytotoxicity similar to that seen in A549. In this study, we mainly used 10 µM of CAR. Lower concentrations of CAR could be used to demonstrate cell survival inhibition and ALDH downregulation, but more studies are warranted to investigate this effect. Given CAR's superior ALDH reduction compared to DEAB, combining CAR with endogenous or exogenous β-agonists could serve as a preclinical or adjuvant lung cancer treatment. Additionally, CAR holds the potential for creating novel, less-toxic derivatives targeting ALDH for further investigation of ALDH's biological role in cancer. Future research should explore NSCLC cell sensitivity to chemotherapy with CAR and β-agonist combined therapy and investigate CAR's antitumor effects across various NSCLC cell lines and cancer types in vitro and in vivo under sympathetic system stimulation.

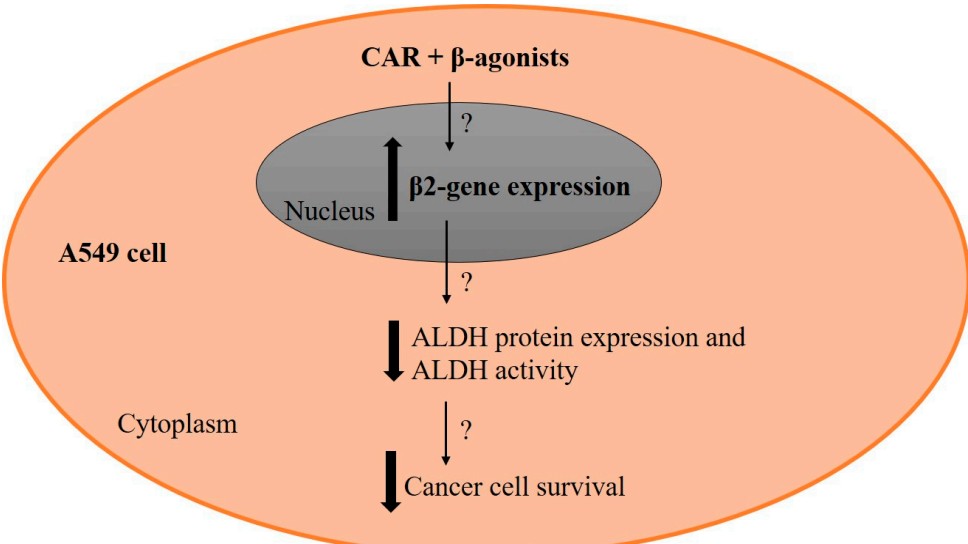

**Figure 7.** Putative mechanism of CAR plus β-agonist in regulating the growth of A549 cancer cells via downregulating ALDH1A1 protein expression and activity. The question mark indicates the unknown mechanism that needs further investigation.

## 4. Materials and Methods

### 4.1. Materials

Human NSCLC cell lines, including A549 (ATCC CCL-185, with LOT numbers 70018877) and H1299 (ATCC CRL-5803, with LOT numbers 70008730) were purchased from the American Type Culture Collection (ATCC, Manassas, VA, USA). Phosphate-buffered saline (PBS), Roswell Park Memorial Institute (RPMI)-1640 medium, trypsin, fetal bovine serum (FBS), and L-glutamine were purchased from Euroclone (Pero, Italy); drugs used

in this research including DEAB, CAR, PROP and isoprenaline hydrochloride (isoproterenol; ISO) were purchased from ACROS organics (Morris Plains, NJ, USA), while EPI was from Sigma (Hertfordshire, UK). 4-Nitrobenzaldehyde (4-NBA), dithiothreitol (DTT), and 3-(4,5-dimethylthiazol-2-yl)-2,5-diphenyl-2H-tetrazolium bromide (MTT), were purchased from Sigma (UK). β-Nicotinamide adenine dinucleotide ($NAD^+$) was purchased from Apollo Scientific (Bredbury, UK). Dimethyl sulfoxide (DMSO) was purchased from TEDIA (Fairfield, OH, USA). RIPA lysis buffer (sc-24948A) was purchased from Santa Cruz Biotechnology (Dallas, TX, USA). Bicinchoninic acid (BCA) assay kit was purchased from Thermoscientific, (Waltham, MA, USA). Western blot reagents were purchased from Sigma (UK), while polyvinylidene difluoride membrane (PVDF) was purchased from Santa Cruz Biotechnology, and primary antibodies anti-ALDH1A1 (ab206396), anti-GAPDH (ab9485) and goat anti-rabbit IgG H&L (HRP) (ab205718) were purchased from Abcam (Cambridge, UK).

### 4.2. Methods

### 4.2.1. Cell Culture

A549 and H1299 cells were cultured in RPMI-1640 medium, supplemented with 10% (*v/v*) FBS and L-glutamine (2 mM) (complete medium). The cells were maintained in a 5% $CO_2$ humidified incubator at 37 °C.

### 4.2.2. Sample Preparation for Measuring ALDH Levels

A549 and H1299 cells were placed in 10 cm dishes and cultured overnight. The next day, cells were subjected to different treatments: CAR at 10 μM, ISO at 20 μM, 40 μM, and 60 μM, EPI at 12.5 μM, 25 μM, and 50 μM, and combinations of CAR at 10 μM with ISO or EPI at the mentioned concentrations. After 120 h, cells were washed with cold PBS, then lysed using RIPA buffer, and the resulting lysates were centrifuged at $13,000 \times g$ for 10 min at 4 °C. This separated proteins in the supernatant from cell debris. The supernatant was stored at $-80$ °C or used directly for measuring ALDH activity and protein expression.

### 4.2.3. Quantification of Total Protein Concentration in Cell Lysate

Total protein concentration in cell lysates was determined using a BCA kit. The protocol followed herein was according to the manufacturer's suggestion. The kit includes a standard protein stock solution with 2 mg/mL concentration. This stock was serially diluted at a 1:2 ratio, resulting in standard protein solutions with known concentrations ranging from 0.125 to 2 mg/mL. A working solution of colorimetric reagents was prepared by mixing 50 parts reagent A with one part reagent B. The reactions in the wells containing standard protein solution and unknown samples started when 200 μL of the working solution was dispensed into each well in a 96-well plate. The plate was left in the incubator at 37 °C for 30 min before measuring absorbance at 562 nm wavelength using the SynergyHTX® spectrophotometer. A standard curve with a straight-line equation was generated by plotting known concentrations of standard protein solutions against absorbance. After determining the unknowns' absorbance, the equation was used to calculate the total protein concentration of unknown samples.

### 4.2.4. NADH Fluorescence Spectrophotometric Activity Assay

The ALDH activity was monitored by measuring the rate of increasing NADH fluorescence signals over a 60 min reaction period. The enzymatic reaction was initiated by the addition of a mixture of the substrate 4-NBA and cofactor $NAD^+$ at a final concentration of 500 μM and 200 μM, respectively, to the wells containing PBS buffer (pH 7.4), a volume of cell lysate (containing 80 μg protein) and reducing agent DTT (5 mM) using a 96-well black plate. The cell lysates were prepared from untreated control and 120 h-treated cells of A549 and H1299 cell lines. The treated samples were prepared as mentioned above. The fluorescence signal of NADH resulting from oxidation of 4-NBA was measured per minute

over 60 min until the plateau was reached through excitation at 340 nm and emission at 460 nm with a SynergyHTX® spectrophotometer at 37 °C. The linear region of the curve was selected to determine the slope. The slope for each cell line under different conditions was calculated from at least five independent experiments. An assay without cell lysate in the reaction mixtures was also carried out with each experiment to confirm the absence of fluorescent signals from any of the mentioned reagents. The results are the percentage of remaining ALDH activity, calculated relative to the untreated control and expressed as the mean with standard error (SE) [85].

### 4.2.5. Western Blot

Samples of A549 and H1299 cells were lysed with RIBA lysis buffer. Total protein concentration was measured using the BCA assay. Approximately 20 μg of isolated proteins was mixed with 5× laemmli loading dye. Proteins were separated by sodium dodecyl sulfate-polyacrylamide gel electrophoresis (SDS-PAGE) (12%) and electrophoretically transferred at a constant current of 25 V for 30 min using a Trans-Blot TurboTM Transfer System (Bio-Rad, Hercules, CA, USA) onto a PVDF membrane. After blocking with 3% *w/v* fat-free milk diluted in TBST (50 mM Tris-HCl; 150 mM NaCl, pH 7.4; 0.1% Tween 20) for 30 min at 4 °C, the membrane was incubated overnight at 4 °C with one of the following primary antibodies: anti-ALDH1A1 or anti-GAPDH. Anti-GAPDH antibody was used as a loading control. The amount of each antibody used was according to the manufacturer's instructions. The next day, after washing, a secondary antibody was applied for 90 min. Then, the membrane was prepared for imaging using a ChemiDocTM Imaging System (Bio-Rad, USA).

### 4.2.6. MTT Cytotoxicity Assay

An MTT assay was conducted as previously outlined [86–88]. Briefly, A549 and H1299 cells were seeded at 750 and 1000 cells per well in 96-well plates and allowed to adhere overnight. Cells were treated with DEAB, β-blockers and β-agonists at concentrations ranging from 0.01 μM to 1000 μM for 96 h. β-blockers used were cardioselective ones (atenolol, esmolol, metoprolol, bisoprolol), non-cardioselective ones (nadolol, propranolol), and a mixed $\alpha/\beta$-blocker (carvedilol) [72,89]. Cell survival was determined using the MTT cell viability assay after 96 h of treatment. MTT solution (50 μL, 0.5 mg/mL) was added and incubated for 3 h at 37 °C, and formazan crystals were dissolved in 200 μL DMSO. Optical densities were measured on a SynergyHTX® spectrophotometer at 540 nm and analyzed with Gen5 Software. The surviving cell fraction relative to the control was calculated from the results, and GraphPad Prism 9.1.0 Software generated dose–response curves from at least three independent experiments. $IC_{50}$ values, representing the drug concentration reducing cell survival by 50% of control, were determined from these curves.

### 4.2.7. Gene Expression Analysis

Gene expression by real-time polymerase chain reaction (PCR) was performed as described before [90]. A549 cells received the same treatments, mentioned in Section 4.2.2.; for 96 h, considering gene rather than protein levels. Total mRNA was extracted from treated and untreated control cells using RNeasy Mini Kit (Qiagen, Germany). Extracted mRNA was quantified, and cDNA was synthesized using a High-Capacity cDNA Reverse Transcription Kit (Thermo Fisher Scientific, Waltham, MA, USA). Gene expression was evaluated by normalizing target genes to ACTB housekeeping gene using a 2-fold change analysis, and calculated relative to the untreated control. Primers from IDT were used, as listed in Table 2. After optimizing conditions, quantitative reverse transcription-PCT (qRT-PCR) was conducted with Advanced SYBR Green Supermix on a Bio-Rad CFX96 real-time PCR system.

**Table 2.** Forward and reverse primer sequences employed for real-time PCR.

| Genes | | Sequence of Nucleotides (5′-3′) |
|---|---|---|
| ALDH1A1 | Forward | CAA GAT CCA GGG CCG TAC AA |
| | Reverse | CAG TGC AGG CCC TAT CTT CC |
| ADRB2 | Forward | CAA GAA TAA GGC CCG GGT GA |
| | Reverse | CCG GTA CCA GTG CAT CTG AA |
| B1AR | Forward | CCG GGA ACA GGA ACA CAC |
| | Reverse | GAA AGC AAA AGG AAA TAT GTC |
| ACTNB | Forward | TTC CTT CCT GGG CAT GGA GT |
| | Reverse | GCA ATG ATC TTG ATC TTC ATT |

4.2.8. Statistical Analysis

Data are shown as individual data points with the mean ± SE represented by a line. GraphPad Prism version 9.1.0 was used for graphing, and data analysis via unpaired *t*-test and one-way ANOVA test. Groups were deemed statistically significant when * $p$-value < 0.05.

**Supplementary Materials:** The following supporting information can be downloaded at: https://www.mdpi.com/article/10.3390/cimb45100505/s1.

**Author Contributions:** B.A.I., A.M.H., H.A. and L.H., Formal Analysis; B.A.I., A.M.H., H.A., L.H., M.A. and R.A., Methodology; B.A.I., Q.M.A. and A.I.M.I., Project Administration; B.A.I., Supervision; B.A.I., Writing—Original Draft Preparation; B.A.I., A.M.H., O.H.A. and L.E., Writing—Review and Editing. All authors have read and agreed to the published version of the manuscript.

**Funding:** This work was financially supported by the Scientific Research Support Fund of the Ministry of Higher Education & Scientific Research (Grant number: MPH/1/55/2019).

**Institutional Review Board Statement:** Not applicable.

**Informed Consent Statement:** Not applicable.

**Data Availability Statement:** Data are contained within the article and within the Supplementary Materials.

**Acknowledgments:** All authors thank the Scientific Research Support Fund of the Ministry of Higher Education & Sci-entific Research for funding this research (Grant number: MPH/1/55/2019). We also thank the Faculty of Pharmacy and Deanship of Scientific Research and Innovation at Al-Zaytoonah University of Jordan for their support.

**Conflicts of Interest:** The authors declare no conflict of interest.

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
