# Peer review of "Investigating Carvedilol’s Repurposing for the Treatment of Non-Small Cell Lung Cancer via Aldehyde Dehydrogenase Activity Modulation in the Presence of β-Adrenergic Agonists"

_cimb, doi:10.3390/cimb45100505_

Round 1

Reviewer 1 Report

Comments and Suggestions for Authors

Ikhmais et al have utilized a β-blocker Carvedilol (CAR) in association with  β-agonist drugs to demonstrate increased lung cancer cell line growth suppression of two NSCLC cell lines A549, and H1299. They show that this growth suppression is due to inhibitory effect on ALDH1 at the protein level and not at the RNA level, a post transcriptional effect. They provide evidence only for A549 as they indicate that ALDH1A1 is not expressed in H1299. Additional convincing evidence is required for acceptance of authors conclusions.

1. Western blot in figure 4A shows equal downregulated ALDH1 A1 expression with 10uM of CAR alone (lane 2) and in the presence of 40 or 60uM of ISO (lanes 7 and 8). Better representative blots are required to show increased effect in the combination treatment.  Lower concentration of CAR could be used in combination with 40 or 60uM ISO that might demonstrate higher downregulation of ALDH1A1 in the combination treatments. Ratio presentation with respect to GAPDH is not a substitute for unambiguous western blot data.

2. There are at least two references in the literature (see below) that shows expression of ALDH or the isozyme ALDH1A2 in the cell line H1299. Expression of these two proteins could be explored with CAR and in combination with ISO or EPI to demonstrate the effect on ALDH.

Lawal B, Kuo YC, Wu AT, Huang HS. Therapeutic potential of EGFR/mTOR/Nf-kb targeting small molecule for the treatment of non-small cell lung cancer.  Am J Cancer Res. 2023 Jun 15;13(6):2598-2616. eCollection 2023.

Jan S Moreb Deniz UcarShuhong HanJohn K AmoryAlex S GoldsteinBlanca OstmarkLung-Ji Chang. The enzymatic activity of human aldehyde dehydrogenases 1A2 and 2 (ALDH1A2 and ALDH2) is detected by Aldefluor, inhibited by diethylaminobenzaldehyde and has significant effects on cell proliferation and drug resistance Chem Biol Interact.. 2012 Jan 5;195(1):52-60.  doi: 10.1016/j.cbi.2011.10.007. Epub 2011 Nov 3. 

3. Authors could try another NSCLC cancer cell line with CAR and  β-agonists if they want to show that ALDH1 (or ALDH1A1) is the target of CAR and  β-agonists.

4. To convince that there is enhanced  β2-adrenergic receptor with CAR and ISO (figure 6C, lane 8), additional cell line studies will be required.

5. A mechanistic figure could be presented to show cell growth regulation by CAR and  β-agonists through ALDH1. 

Reviewer 2 Report

Comments and Suggestions for Authors

In this study, Ikhmais et al evaluated the role of the anticancer potential of a subset of β-blockers in NSCLC cell lines; A549 and H1299. This is an interesting topic, but there are some potential points that can improve the content of the paper;

1) The authors should mention different lung cancer treatments in the introduction, before line 49, using relevant publications such as Jianwei Zhu, et al - 2021 - Monireh Mohsenzadegan, et al. - 2020, Fausto Petrelli, et al - 2021 and then conclude why they selected these drugs.

2) The expression levels of ALDH1 in two cell lines should be discussed with other relevant publications such as Ali Samadikuchaksaraei, et al - 2014 and so on...

3) The clinical significance of these data should be added to the paper. 

Comments on the Quality of English Language

minor English edit 
